

# Comparative evaluation of machine learning algorithms for phishing site detection

Noura Fahad Almujahid[1,*], Mohd Anul Haq[2,*] and Mohammed Alshehri[1]

[1] Department of Information Technology, College of Computer and Information Science, Majmaah University, Majmaah, Riyadh, Saudi Arabia
[2] Department of Computer Science, College of Computer and Information Sciences, Majmaah University, Al Majmaah, Riyadh, Saudi Arabia
[*] These authors contributed equally to this work.

Corresponding author
Mohd Anul Haq, m.anul@mu.edu.sa

## ABSTRACT

The advent of Internet technologies has resulted in the proliferation of electronic trading and the use of the Internet for electronic transactions, leading to a rise in unauthorized access to sensitive user information and the depletion of resources for enterprises. As a consequence, there has been a marked increase in phishing, which is now considered one of the most common types of online theft. Phishing attacks are typically directed towards obtaining confidential information, such as login credentials for online banking platforms and sensitive systems. The primary objective of such attacks is to acquire specific personal information to either use for financial gain or commit identity theft. Recent studies have been conducted to combat phishing attacks by examining domain characteristics such as website addresses, content on websites, and combinations of both approaches for the website and its source code. However, businesses require more effective anti-phishing technologies to identify phishing URLs and safeguard their users. The present research aims to evaluate the effectiveness of eight machine learning (ML) and deep learning (DL) algorithms, including support vector machine (SVM), k-nearest neighbors (KNN), random forest (RF), Decision Tree (DT), Extreme Gradient Boosting (XGBoost), logistic regression (LR), convolutional neural network (CNN), and DL model and assess their performances in identifying phishing. This study utilizes two real datasets, Mendeley and UCI, employing performance metrics such as accuracy, precision, recall, false positive rate (FPR), and F-1 score. Notably, CNN exhibits superior accuracy, emphasizing its efficacy. Contributions include using purpose-specific datasets, meticulous feature engineering, introducing SMOTE for class imbalance, incorporating the novel CNN model, and rigorous hyperparameter tuning. The study demonstrates consistent model performance across both datasets, highlighting stability and reliability.

# INTRODUCTION

Phishing is a method that aims to use technological and social tricks to gain access to customers' financial and personal information. Social media platforms employ spoofed

emails from well-known businesses and organizations, leading users to fraudulent websites where they are prompted to disclose sensitive information such as financial details, usernames, and passwords. Hackers employ malicious programs to steal credentials, often intercepting usernames and passwords from users' online accounts. Phishers employ various methods to steal user information, including phone calls, text messages, instant messaging, uniform resource locator (URL), instant chat, and forum postings. Phishing attempts are designed to mimic legitimate content, tricking individuals into divulging sensitive information. The primary objective of phishing is to acquire personal data for identity theft or financial gain. Around the world, phishing attacks cause significant economic harm. Furthermore, Following the latest Phishing pattern research from the Anti-Phishing Working Group "APWG" (*Tally et al., 2023*).

Most phishing attacks target webmail and financial/payment institutions. Criminals create unauthorized reproductions of legitimate websites and emails, typically from an organization that deals with financial information, to get private information (*Jain & Gupta, 2018*; *Giri et al., 2021*; *Purbay & Kumar, 2021*). This email is designed with the logos and phrases of a reputable firm. HTML's architecture and structure allow copying pictures or an entire web page (*Le et al., 2018*). It is also a key element contributing to the Internet's quick development as a tool for communication. It permits the abuse of brand names, logos, and other corporation identifiers that customers depend on for authentication measures (*Hong et al., 2020*; *Abutair & Belghith, 2017*; *Kumar et al., 2020*) to trick users. As many "spooled" emails as possible are sent by the phisher. Upon opening these emails, clients are often directed away from the actual company and onto a phony website user information is subject to a high risk of exploitation. Because of these factors, phishing is important however difficult, and vital in modern culture (*Rao & Pais, 2019*; *Aljofey et al., 2020*).

As per the cyber security authority of Saudi Arabia, the recent phishing attacks in Saudi Arabia on April 7, 2021, increased by about 300%, and the importance of awareness and prevention of phishing attacks (*Alharbi et al., 2022*). Several recent studies have been conducted against phishing based on domain features like web pages, and website content, incorporating the URLs and the website's content, the website's source code, and a snapshot of the website (*AlEroud & Karabatis, 2020*). Most current studies are limited to URL content and emails only, and there are no solutions to detect phishing in text messages or social media such as WhatsApp, Twitter, LinkedIn, and others. However, there is a need for more appropriate anti-phishing technologies in a business to detect phishing URLs and safeguard its users. There are many ways to detect phishing URLs, such as ML and blacklisting. It is possible to find malicious URLs on the Internet. Recognized using ML techniques (*Gupta & Rani, 2020*; *Joshi & Pattanshetti, 2019*). A blacklist is a key component of the traditional URL detection technique, which is a list of malicious URLs gathered from user reports or professional judgment. The URL in the blacklist is frequently updated, and on the one hand, a URL is verified using the blacklist. However, there are an increasing number of harmful URLs that are not yet on the blacklist. Cybercriminals, for instance, can create fresh malicious URLs using a domain generation algorithm to get around the blacklist.

As a result, It is incredibly difficult to recognize malicious URLs using an exhaustive blacklist (*Wu, Kuo & Yang, 2019*; *Chiew et al., 2015*).

In contrast to many earlier approaches, Researchers concentrate on finding malicious URLs among a variety of URLs. The objective of this article is to conduct an empirical evaluation of ML algorithms for phishing detection. The novelty of the present investigation lies in several key aspects. Firstly, it introduces a comparative analysis of ML and DL algorithms using two distinct datasets, Mendeley and UCI. This diversification in dataset sources enhances the generalizability of the findings. Furthermore, the study addresses class imbalance in the datasets by employing the oversampling technique. Notably, the novel CNN model was developed which showcased significant improvements in accuracy compared to previous studies, while addressing the gaps in phishing analysis and detection through data preprocessing and hyperparameter tuning, ultimately enhancing computer system security, and providing valuable insights for future research.

## BACKGROUND AND RELATED WORK

The classification of phishing attacks is carried out according to the attacker's mechanism for deceiving users. Some forms of such attacks are keyloggers, DNS toxicity, social engineering blog operations, messaging services (SMS), social media platforms such as Twitter and Facebook, and file-sharing services, *etc.* (*Jain & Gupta, 2018*). Each form of phishing has slight differences in the way the process is carried out to deceive the user. Phishing attacks occur *via* email or SMS, where these messages contain a link to direct the user to phishing sites. Phishing detection using ML is a burgeoning subject of study with an increased desire to use deep learning (DL) methods.

*Le et al. (2018)* proposed URL Net, a deep neural URL detection network based on CNN. They asserted that current approaches, which frequently employ Bag of Words (BoW) style features, have certain critical flaws, including the inability to recognize sequential ideas in URL strings, a failure to detect real-time URLs containing hidden features, and the lack of automated feature extraction. The network was built, and CNNs and Word CNNs for characters were produced. Additionally, they offered sophisticated methods that were especially useful for dealing with rare phrases, a challenge that frequently arises in malicious URL identification activities. Using this strategy, URLNet can recognize URLs during the testing phase by utilizing embeddings and subword data from hidden words. Another study by *Abutair & Belghith (2017)*, proposed a URL detector that can identify phishing attempts. They contended that the approach could be purposefully and scaled modified to fit different sizes. They collected 572 cases for both trustworthy and malicious URLs, and the traits were extracted and weighed for use in the prediction process. The test results were trustworthy both in the presence and absence of online phishing threats. The genetic algorithm (GA) was used to improve accuracy.

*Kumar et al. (2020)* looked into how accurately phishing URLs can be distinguished from benign URLs in a collection of URLs. They discussed statistical analysis, host-based lexical analysis, feature engineering, randomization, and feature extraction. Multiple classifiers were used for the comparative study, and it was discovered that the outcomes were

broadly consistent. The authors claimed that their approach was practical for removing functionality from URLs using short common words. Additional features that produce the best results could be tested. Some older URLs can be found in the study's dataset, which may result in a chance of underperformance.

*Rao & Pais (2019)* used a parameter that compares the similarity between the suspect site and the corresponding domain and achieved 98.61% accuracy, a 97.77% true positive rate, and a false positive rate of less than 0.64% according to the experimental results. *Aljofey et al. (2020)* suggested using a convolutional neural network (CNN) to recognize phishing URLs. To collect the URL data for this study, researchers used a sequential sequence. On benchmark datasets, it obtained accuracies of 98.58%, 95.46%, and 95.22%, respectively.

*Yerima & Alzaylaee (2020)* suggested conducting experiments to evaluate CNN models. Python was used to implement the models, and both the TensorFlow backend and the Keras library were employed. Additionally, Pandas, ScikitLearn, Seaborn, and NumPy were used. The dataset included 11,055 instances collected from 6,157 trusted websites and 4,898 phishing websites. They found that the CNN2 model performs better with more filters. The highest accuracy was obtained when 64 filters were used, with an $F1$-score of 0.974, contrasting accuracy of 59.8% with an $F1$-score of 0.963 when using only eight filters.

*Mohammad, Thabtah & McCluskey (2014)* built their model using seventeen characteristics gathered from URLs and the source code of 600 legitimate and 800 fraudulent websites. They employed the "hold-out" validation approach to avoid the problem of overfitting by splitting their datasets into testing, validation, and training datasets. They used the "log sigmoid" activation function. *Khan & Rana (2021)* suggest detecting malicious URLs using minimal features, consisting of feature extraction and classification techniques. The experiments were conducted using a dataset consisting of 3,000 cases. Accuracy and error rates were used as computational measures, respectively. The DNN method was used to verify the correctness of the feature selection. The researchers found that the individual accuracy rate of the experiment ranges from 61.06% to 97.07%. Two URL-based features for which the accuracy rate was less than 66 were separated, resulting in a test accuracy of 99.13% and a training accuracy of 99.71%. Thus, the DNN's training accuracy was 99.90%.

*Dunlop, Groat & Shelly (2010)* used the FishTank database and 100 phishing sites. They applied the concept of using optical character recognition to turn logos and screenshots of images into text, thereby reducing the approach of queries to a single query. *Varshney, Misra & Atrey (2016)* used page titles and URLs only to build a powerful search string specifically pinpointing phishing websites. They developed a working prototype for Google Chrome as a benchmark (LPD). The authors suggested adding additional features to upcoming work while maintaining resource efficiency, which is the main idea of the LPD proposal. *Jain & Gupta (2018)* presented a technique for URL-based anti-phishing using machine learning. To verify the effectiveness of their strategy, they used 14 characteristics from the URL to determine whether a website is legitimate or malicious. The recommended approach was trained using over 33,000 phishing and legitimate internet sites for SVM and NB classifiers. The process of learning was the main emphasis of the phishing detection approach. They

identified 14 distinct characteristics that distinguish authentic websites from phishing ones. When SVM classification is used, the results of their trial have above 90% accuracy.

*Nguyen et al. (2013)* identified six minimal features and claimed to provide high accuracy. They used the 11,660 phishing sites in the Phishtank database with an accuracy of 97.16%. Their operations heavily rely on third parties. *Ramesh, Krishnamurthi & Kumar (2014)* suggested using DNS searches and the precise target domain for matching *via* links from HTML sources. The Phishtank database served as the dataset with an accuracy of 99.62%. *Singh, Maravi & Sharma (2015)* tested Madaline and backpropagation for phishing website classification using neural network training on top of SVM with over 15 features. They claimed that Adaline's classification of 179 phished websites from the Phishtank database was more accurate and effective. Alexa's accuracy rate for 179 legitimate websites is 99.14%. Table S1. shows a comparison of the literature review.

## METHODOLOGY

### Dataset description

The present investigation used two datasets (*Brownlee, 2021*; *Samad et al., 2023*) for phishing site detection. The primary dataset, sourced from the Mendeley repository (*Brownlee, 2021*), consisted of 48 features extracted from a collection of 10,000 web pages. Among these, 5,000 were identified as phishing sites, while the remaining 5,000 were verified as legitimate websites, see Fig. S1. To compile the list of legitimate websites, Alexa and Common Crawl were utilized, whereas PhishTank and Open-Phish were employed to compile the list of malicious sites (*Brownlee, 2021*). By examining the extraction of content and URL features, we can achieve high-performance phishing detection. Additionally, it is crucial to determine the usefulness of deep classification for this task and whether converters are necessary for full-text analysis to identify the appropriate features. The dataset provides four types of features that can be extracted for predicting phishing based on the URL. Address bar-based features and abnormal features are presented in Tables S2 and S3, respectively.

The second dataset employed in this study was sourced from the UCI Repository (*Samad et al., 2023*), comprising 11,055 records with 31 features. Among these, 11,055 sites, and 4,898 sites were identified as phishing sites (labeled as −1), while the remaining 6,157 were verified as legitimate websites (labeled as 1). The dataset contains 31 features, 30 are independent, while 1 serves as the target variable categorized into four groups.

The decision to use the specified datasets for phishing site detection was deliberate and strategic. These datasets were chosen due to their direct relevance to the research objectives, accessibility, quality, and size. With substantial instances and diverse features, they provide a solid foundation for training and evaluating machine learning and deep learning models. While there are other relevant datasets (*Dunlop, Groat & Shelly, 2010*; *Nguyen et al., 2013*), these were deemed the most suitable for achieving the study's goals efficiently and effectively.

## Data preprocessing and feature engineering

The present investigation used libraries including matplotlib, seaborn, pandas, and NumPy for data pre-processing. The dataset contains features and labels as per Tables S4 and S5. Out of 48 features in Dataset 1, The HttpsInHostname feature has no use in the case of Dataset 1. The datasets suggest legitimate, and phishing as 1, and 0, respectively, with 5,000 instances each, see Fig. S1. Dataset 1 was balanced for both the classes.

The first feature index has no use in the case of Dataset 2; therefore, it was dropped. The datasets suggest legitimate, and phishing as 1, $-1$, respectively. The number of occurrences for legitimate and phishing were 6,157 and 4,898, respectively Fig. S2.

Dataset 2 was not well balanced for both the classes; therefore, oversampling the minority class was performed, see Fig. S2. The Synthetic Minority Over-sampling Technique (SMOTE) was used for oversampling the minority class. The present investigation employed SMOTE with the auto-sampling strategy to address class imbalance in the dataset. The decision to use auto was based on the algorithm's capability to dynamically determine suitable oversampling ratios for each class. This approach accommodates varying degrees of class imbalance without necessitating a predefined fixed ratio, allowing for adaptability to the dataset's specific characteristics. The auto strategy aligns with a data-driven and flexible methodology, enabling the algorithm to autonomously adjust to the observed distribution of classes in the dataset. The tuning for the k_neighbors parameter was implemented in the SMOTE algorithm. It iterated through different values of k_neighbors (3, 5, 7, and 9), applied SMOTE to the training data for each iteration, and trained the classifier on the resampled data. The accuracy of the classifier is then assessed on the resampled test set for each k_neighbors value. It was found that $K = 5$ provided optimal results in terms of accuracy and achievement of the class balance.

After applying SMOTE, a balanced class distribution was achieved for Dataset 2, see Fig. S2. Each class (represented by $-1$ and 1) had 6,157 instances, effectively addressing the class imbalance issue. A balanced distribution was considered beneficial for machine learning models as it ensured that the model was exposed to a similar number of examples from each class during training.

Analyzing correlations is a crucial aspect of EDA. Primarily, correlation analysis allows us to gauge the strength and direction of the association between two variables. A positive correlation signifies that as one variable increases, the other tends to increase as well, while a negative correlation indicates an inverse relationship. This insight is crucial for understanding how variables interact within the dataset. Additionally, correlation analysis is instrumental in uncovering patterns and trends in the data. Identifying relationships between variables can reveal dependencies and guide further investigation into the underlying dynamics of the dataset. Furthermore, correlation analysis aids in feature selection for modeling purposes. Highly correlated features may carry redundant information, and identifying and excluding such features can streamline the model, enhancing its interpretability and performance. Hence, a correlation analysis was conducted to examine the relationships among data features (see Figs. S3 and S4).

The correlation analysis for Dataset 1 revealed the notable correlations within the dataset, the top correlation (0.8730) is observed between 'NumQueryComponents' and

'NumAmpersand', indicating a strong positive relationship. The second-highest correlation (0.8118) exists between 'QueryLength' and 'NumQueryComponents', signifying a substantial positive correlation. Additionally, the third-ranking correlation (0.7544) is identified between 'QueryLength' and 'NumAmpersand', representing a noteworthy positive association. The fourth-highest correlation (0.6492) is noted between 'UrlLength' and 'QueryLength', revealing a moderately positive correlation between these feature pairs. The analysis for Dataset 2 revealed a strong correlation between the favicon and popup window features, suggesting that websites obtaining favicon from external sources often dominate the text field within the pop-up window. Moreover, the SSL certificate final stage and URL of the anchor exhibited a notable correlation with the likelihood of phishing. To represent phishing, labels with a value of −1 were transformed to 0, while labels with a value of 1 denoted non-phishing instances.

Feature engineering involves creating new features or transforming existing ones to improve model performance or extract useful information from the data. Feature engineering was performed by calculating Theil's uncertainty coefficient (TU) and Point Biserial Correlation Coefficient (PBCC). The TU measures the predictability of the target variable given each categorical feature. By leveraging these techniques, the analysis identified the most relevant numerical and categorical features correlated with the target variable. Similarly, The PBCC quantifies the linear relationship between each numerical feature and a binary target variable. For Dataset 1, the first step towards feature engineering was to segregate the target variable ('CLASS_LABEL') and ID from the dataset. Then, the categorical and numerical features were separated, finding 29 categorical features and 19 numerical features. Subsequently, it calculates the TU for each categorical feature, revealing their correlation with the target variable. The top correlated categorical features, such as 'PctExtNullSelfRedirectHyperlinksRT', 'FrequentDomainNameMismatch', 'ExtMetaScriptLinkRT' *etc.*, are filtered and converted back to the integer type. For the numerical features, the PBCC was computed. The top correlated numerical features, such as 'NumDash', 'PctNullSelfRedirectHyperlinks', 'NumDots', *etc.*, are filtered. Finally, the 13 filtered categorical and numerical data features with high scores were merged with the target variable, see Table S6. For Dataset 2 the index and the target variable (Result) were segregated and the scores of the features were calculated similarly to Dataset 1. The SSLfinal_State showed the significantly highest value of 0.715, followed by URL_of_Anchor with a value of 0.693. The 11 filtered features with high scores were merged with the target variable, see Table S6.

## Machine learning and deep learning models

In the present investigation several popular ML techniques, including SVM, KNN, RF, DT, XGBoost, LR, and CNN were employed to assess their accuracy in identifying phishing sites using two real datasets. To ensure a reliable evaluation, k-fold cross-validation was utilized. The dataset was divided into k equal-sized folds, where k−1 folds were used for training, and the remaining fold was used for testing. In this experiment, a value of $k = 5$ was set initially. Out of the total 48 features, the SelectKBest feature extraction technique was employed to select the most informative 30 features for classification in this study. SelectKBest ranks the

features based on their statistical significance and selects the top K features. By using this approach, each fold was utilized for testing, and the average accuracy across all folds was computed, providing a more robust measure of the ML models' performance. To prevent overfitting, an additional step was taken during the hyperparameter tuning process using GridSearchCV.

## Logistic regression (LR)

In the first step of the analysis, LR was employed, which is commonly used for predictive analytics and classification tasks. LR calculates the likelihood of an event occurring based on a given dataset of independent variables. In this approach, the dependent variable ranges from 0 to 1, representing the outcome as a probability. To transform the odds, which is the probability of success divided by the probability of failure, the logit formula was utilized, as shown in Eqs. (1) and (2).

$$\text{Logit}(P) = \frac{1}{1 + \exp(-p)}. \tag{1}$$

The logistic function, Logit ($p$), transforms a linear combination of features into a probability range (0–1):

$$\text{In}\left(\frac{p}{1-p}\right) = \beta_0 + \beta_1 X_1 + \cdots + \beta_k X_k. \tag{2}$$

where $In$ is the natural logarithm. $p$ is the probability of an event. $X_1, X_2, X_k$ are predictor variables. $\beta 0, \beta 1, \ldots, \beta k$ are coefficients.

## K-nearest neighbors (KNN)

The KNN algorithm is a supervised learning classifier that uses proximity to classify or predict the grouping of a single data point. It can be applied to both classification and regression issues. KNN works by measuring the similarity between query points and other data points based on their distance or closeness. Euclidean distance is one of the commonly used methods for calculating distance, as shown in Eq. (3). Euclidean distance measures the straight line between the query point and the available point. While KNN is easy to use and adaptable, it suffers from memory and overfitting issues. An instance of the K-NeighborsClassifier class is created with the initial number of neighbors (K) set to 5. The number of K was tuned using GridSearchCV to prevent the overfitting issue.

$$d(x, y) = \sqrt{\sum_{i=1}^{n} (y_i - x_i)^2}. \tag{3}$$

where $d(x, y)$: This represents the Euclidean distance between points $x$ and $y$. $n$: The number of dimensions or features in the dataset. $yi$: The $i$th component of point $y$. $xi$: The $i$th component of point $x$.

## Decision tree (DT)

DT is a non-parametric supervised learning approach used for both classification and regression applications. Its hierarchical tree structure consists of a root node, branches,

internal nodes, and leaf nodes. To find the best-split points inside a tree, decision tree learning uses a greedy search method, a divide-and-conquer tactic. The dividing procedure is then repeated top-down and recursively until all or most records have been assigned to certain class labels. The complexity of the decision tree significantly affects whether all data points are categorized as homogeneous sets. Smaller trees are more likely to attain pure leaf nodes, meaning a single class of data items. To prevent overfitting the DT model was optimized using GridSearchCV, for the parameters including criterion, max_depth, and min_samples_split.

## Random forest (RF)

The random forest method builds each decision tree in the ensemble from a data sample taken from a bootstrap sample. The random forest algorithm extends the bagging technique, which produces a nonstationary forest of decision trees using feature randomness in addition to bagging. Feature randomness ensures low correlation across decision trees and creates a random collection of features. Random forests merely choose a portion of those feature splits, whereas decision trees consider all potential feature splits. The hyperparameters tuned using GridSearchCV included the number of trees, maximum tree depth, and minimum number of samples required for node splitting.

## Support vector machines (SVM)

SVM is a reliable classification and regression method that increases a model's predicted accuracy while preventing overfitting on the training set. SVM is particularly well-suited for data analysis with a very large number of predictor fields, such as thousands. SVM categorizes data points even when they are not linearly separable by mapping the data to a high-dimensional subspace. Once a divider between the classes is identified, the data are converted to enable the hyperplane representation of the separator. By carefully adjusting the hyperparameters, such as the regularization parameter (C), kernel type, and kernel coefficient (gamma), the SVM model aimed to strike a balance between model complexity and the ability to generalize well to unseen data.

## XGBoost

XGBoost is a gradient-boosted decision tree implementation created for speed and performance. It is implemented through the XGBoost package. Gradient boosting decision tree implementation is done *via* this package. Boosting is an ensemble technique where new models are taught from the errors of older ones. Models are gradually introduced until no further advancements are possible. The AdaBoost method is a well-known example that weights data points that are challenging to forecast. XGBoost supports both regression and classification. The XGBoost model was optimized using GridSearchCV, for the parameters including criterion, max_depth, and min_samples_split.

## CNN model

In this investigation, we developed a robust and sophisticated predictive model for phishing detection, employing a CNN architecture. The details of the tabular data were systematically addressed through the incorporation of multiple convolutional and pooling layers within

the model. These architectural components were thoroughly designed to extract nuanced patterns and relationships present in the dataset. The convolutional layers, featuring increasing filter sizes, and the strategic integration of max-pooling layers for down-sampling collectively contributed to the model's ability to recognize subtle nuances in the tabular input. Additionally, densely connected layers were introduced, accompanied by dropout regularization, strategically applied to mitigate the risk of overfitting. The output layer, characterized by a sigmoid activation function, facilitated binary classification, effectively distinguishing between legitimate and phishing websites.

Furthermore, the architectural details of the developed CNN model were visually represented using Fig. S5. Figure S5 illustrates the connectivity and structural attributes of each layer within the network. The training process unfolded over 10 epochs, utilizing a batch size of 32, and comprehensive evaluations were conducted using a suite of performance metrics, including accuracy, precision, recall, and $F1$ score.

### Deep learning model

In the present investigation, a DL model was developed for phishing detection using two datasets. The model architecture was constructed using TensorFlow's Keras API, comprising three dense layers, each followed by a dropout layer for regularization, Fig. S6. The first dense layer had 64 units and utilized the rectified linear unit (ReLU) activation function. Subsequently, a dropout layer randomly sets a fraction of input units to zero to prevent overfitting. The second dense layer had 32 units and also used the ReLU activation function, followed by another dropout layer. The final dense layer consisted of a single unit with a sigmoid activation function, suitable for binary classification tasks. After defining the model, it was compiled using the Adam optimizer with a binary cross-entropy loss function, common for binary classification problems. Additionally, accuracy, precision, recall, and $F1$ scores were chosen as the evaluation metrics for model performance during training. The model was then trained using the training data for 20 epochs with a batch size of 32, while also validating a portion of the training data to monitor performance and prevent overfitting. The training process yielded a history object containing information about the training process.

### Hyperparameters tuning

In this section, the performance of the ML models with hyperparameter tuning using gridsearchCV is analyzed and compared in terms of accuracy, precision, *etc.*, see Table S7.

The study exclusively utilized grid search cross-validation (GridSearchCV) for hyperparameter tuning due to its simplicity, effectiveness, and thorough exploration of the hyperparameter space. This method systematically evaluates all combinations within a predefined grid, ensuring comprehensive tuning and robust model performance. Without hyperparameter tuning, the model may not achieve its maximum potential, resulting in suboptimal performance. GridSearchCV's straightforward implementation and interpretability make it suitable for this research, enabling the attainment of optimal results.

For LR, L2 penalty, C at 0.1, Saga solver, and 500 max iterations were optimal. The DT model favored the gini criterion, max depth of 3, and min samples leaf of 5. Random forest

excelled with 150 estimators, max depth of 10, min samples split of 5, min samples leaf of 2, and Log2 for max features. K-NN leaned towards three neighbors and brute algorithm. SVC tuned to C 0.7 and the Sigmoid kernel. XGBoost chose a learning rate of 0.2, 100 estimators, max depth of 5, min child weight of 2, subsample of 0.8, and colsample bytree of 1.0. The CNN featured 64 filters, (3, 3) filter size, (3, 3) pool size, 128 dense neurons, and a 0.5 dropout rate. For the deep learning model, hyperparameters such as optimizer, learning rate, batch size, and dropout rate were tuned to optimize model performance. These settings are aimed at enhancing model predictive capabilities, considering algorithmic nuances and dataset intricacies.

### Experimental design

The D1 contains 10,000 instances while the D2 contains 12,314 instances with class balance in both datasets. In the present investigation, a Stratified K-Fold cross-validation method with 10 splits, was employed to enhance the robustness of the model evaluation process. In this study, we used Google's specialized processors called Tensor Processing Units (TPUs) v2–8. These TPUs speed up the training of AI models. The TPU v2–8 had eight cores and 64 GiB of memory. On average, the CNN model took 94 s and 29 ms to complete the training cycles, while ML models took less time, which was under 10 s for all the models.

## EVALUATION MEASURES

In this section, we evaluate the resulting effectiveness of seven ML and DL models using four measures, namely precision, recall, $f$1-score, accuracy, and false positive rate (FPR) for analyzing the results. The FPR measures the proportion of actual negatives incorrectly classified as positives by a model, indicating its ability to avoid false alarms. Accuracy means the ratio of the number of web pages detected as phishing pages to the number of total regular web pages. The recall is the ratio of the number of web pages detected as phishing pages to the number of total phishing samples. precision is the ratio of the number of pages detected as phishing pages to the total detected web pages. Accuracy, recall, precision, and FPR are calculated in Eqs. (4), (5), (6) and (7) (*Haq, 2022*).

$$Accuracy = \frac{TP + TN}{TP + FP + TN + FN} \tag{4}$$

$$Precision = \frac{TP}{TP + FP} \tag{5}$$

$$Recall = \frac{TP}{TP + FN} \tag{6}$$

$$FPR = \frac{FP}{FP + TN}. \tag{7}$$

The number of classified phishing pages is referred to as the true positive (TP). True negative (TN) is the number of legitimate pages that have been correctly classified.

The number of phishing pages misclassified as legitimate pages is referred to as the false negative (FN). The number of legitimate pages misclassified as phishing pages is referred to as false positives (FP). Furthermore, we use the $F$1-score in Eq. (8) as a metric to evaluate

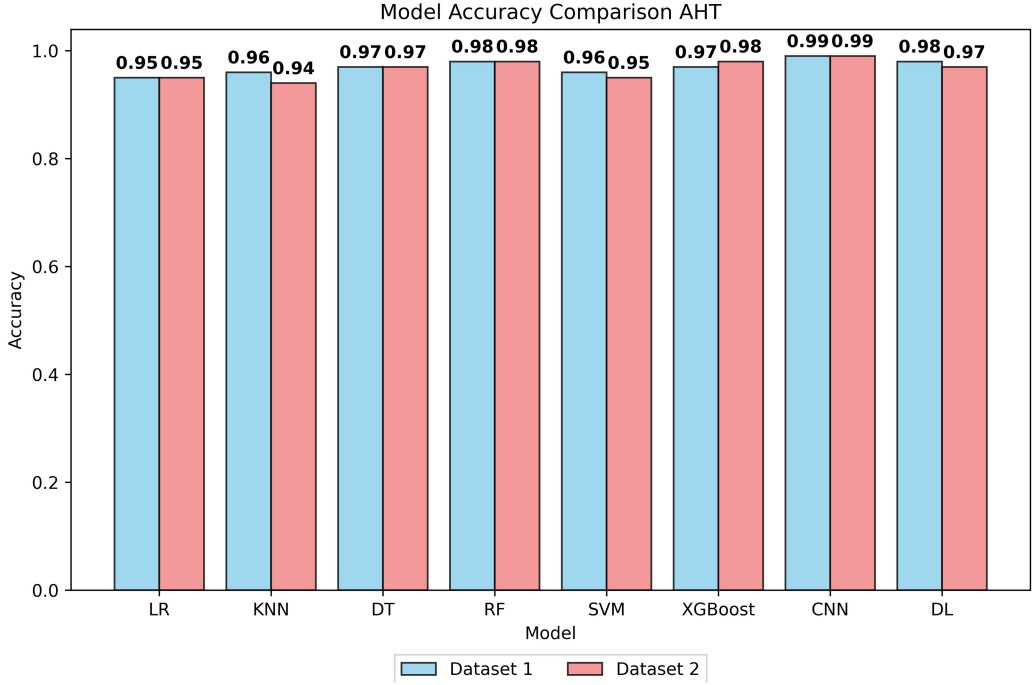

Figure 1 **Accuracy of all models after hyperparameter tuning on both datasets.**

our approach.

$$F1 = \frac{\text{Precision} + \text{Recall}}{\text{Precision} \times \text{Recall}}. \tag{8}$$

# RESULTS AND ANALYSIS

The performance of eight classification algorithms was evaluated using five metrics, namely precision, recall, $f1$-score, FPR, and accuracy. In evaluating ML models on Mendeley (Dataset 1) and UCI (Dataset 2), diverse algorithms including LR, KNN, DT, RF, SVM, XGBoost, CNN, and DL were assessed for their inherent capabilities before hyperparameter tuning (BHT), and after hyperparameter tuning (AHT), see Fig. S7, Fig. 1 and Table S8.

The LR model demonstrated an accuracy of 94% at BHT on Dataset 1, improving marginally to 95% at AHT, with stable precision, recall, and $F1$-score values. On Dataset 2, consistent performance was observed with an accuracy of 93% at BHT and 94% at AHT. For the KNN model, an accuracy of 95% was achieved on Dataset 1 at both BHT and AHT, with stable precision, recall, and $F1$ scores. On Dataset 2, the model attained an accuracy of 94% at BHT and maintained 94% accuracy at AHT. The DT model showcased high accuracy across both datasets, with BHT accuracies of 97% on Dataset 1 and 96% on Dataset 2. Post-tuning, accuracy remained high at 97% and 96%, respectively. The SVM model displayed accuracies of 95% and 96% at BHT on Dataset 1 and 94% on Dataset 2. After AHT, accuracies were maintained, with precision showing slight improvement.

RF model accuracies were consistently high at 97% on Dataset 1 and 96% on Dataset 2, with minimal variation post-tuning. XGBoost model exhibited accuracies of 93% and 98% at BHT and AHT on Dataset 1, and 91% and 98% on dataset 2. Performance remained stable across both datasets. The DL model achieved accuracies of 95% and 98% at BHT and AHT on Dataset 1, with consistent precision, recall, and $F1$ scores. The CNN model outperformed others with accuracies of 97% and 99% at BHT and AHT on Dataset 1, and 95% and 99% on Dataset 2, with stable precision, recall, and $F1$-scores.

All models demonstrated low FPR values, indicating effective data preprocessing and hyperparameter tuning. CNN's superior performance is attributed to its feature extraction capability, capturing intricate patterns and spatial hierarchies in the dataset. The better performance of the CNN model in the present study is due to its natural ability to automatically extract relevant features from the input data, reducing the need for manual feature engineering. This feature extraction capability allows them to adapt and generalize well to diverse and complex datasets. The findings of the present investigation highlighted that CNNs excel because of their ability to capture intricate patterns and spatial hierarchies in the dataset.

## COMPARISON WITH OTHER STUDIES

In comparison to existing studies, our research stands out through distinctive elements. Firstly, our study introduces a comparative analysis of ML and DL algorithms, utilizing two real datasets, Mendeley and UCI. This deliberate choice enhances the robustness and generalizability of our findings, setting this study apart. Particularly noteworthy is the superior performance of the CNN model in intrusion detection, a contribution highlighted in our results. This unique insight into CNN's efficacy represents a significant advancement compared to previous works. Secondly, the present investigation used meticulous feature engineering for both datasets using TU and PBCC techniques. Additionally, the present study addresses class imbalance in Dataset 2 through the application of SMOTE. By incorporating purpose-specific datasets and employing rigorous hyperparameter tuning using the GridSearchcv approach, this research significantly enriches the experimental scope, distinguishing itself as a valuable contribution to the field. The study demonstrates consistent model performance across both datasets, highlighting the stability and reliability of the proposed models. Table 1 presents the comparison with other studies for all the models at the AHT phase.

The LR model in our study demonstrates superior accuracy, achieving 95% compared to the 93% reported in *Samad et al. (2023)*. This notable difference primarily stems from our extensive hyperparameter tuning. Our approach involved exploring a wider range of hyperparameters such as penalty, C, solver, and max_iter through gridsearchCV, whereas (*Samad et al., 2023*) employed fewer combinations. Similarly, subtle variations in the performance of other models can also be attributed to rigorous hyperparameter tuning.

Comparisons with *Haq (2022)* reveal consistent trends, with models like RF and XGBoost performing well across datasets. Interestingly, *Alsharaiah et al. (2023)* introduces variability in KNN and gradient-boosting performance, emphasizing the influence of

**Table 1** Comparison with the other studies for all the models after the AHT phase.

| Studies | Accuracy | Precision | Recall | F1-score | Reference | Dataset |
|---------|----------|-----------|--------|----------|-----------|---------|
| LR | 0.95 | 0.94 | 0.94 | 0.94 | Current Study | 1 |
| KNN | 0.96 | 0.94 | 0.95 | 0.94 | Current Study | 1 |
| DT | 0.97 | 0.97 | 0.97 | 0.97 | Current Study | 1 |
| RF | 0.98 | 0.98 | 0.97 | 0.98 | Current Study | 1 |
| SVM | 0.96 | 0.96 | 0.95 | 0.95 | Current Study | 1 |
| XGBoost | 0.97 | 0.96 | 0.95 | 0.96 | Current Study | 1 |
| CNN | 0.99 | 0.98 | 0.98 | 0.99 | Current Study | 1 |
| DL | 0.98 | 0.98 | 0.97 | 0.97 | Current Study | 1 |
| LR | 0.95 | 0.94 | 0.94 | 0.94 | Current Study | 2 |
| KNN | 0.94 | 0.95 | 0.95 | 0.94 | Current Study | 2 |
| DT | 0.97 | 0.97 | 0.96 | 0.96 | Current Study | 2 |
| RF | 0.98 | 0.97 | 0.97 | 0.97 | Current Study | 2 |
| SVM | 0.95 | 0.95 | 0.95 | 0.94 | Current Study | 2 |
| XGBoost | 0.98 | 0.97 | 0.98 | 0.97 | Current Study | 2 |
| CNN | 0.99 | 0.99 | 0.98 | 0.99 | Current Study | 2 |
| DL | 0.99 | 0.98 | 0.98 | 0.98 | Current Study | 2 |
| LR | 0.93 | 0.92 | 0.95 | 0.93 | *Samad et al. (2023)* | 1 |
| SVM | 0.94 | 0.94 | 0.94 | 0.94 | *Samad et al. (2023)* | 1 |
| NB | 0.84 | 0.94 | 0.72 | 0.81 | *Samad et al. (2023)* | 1 |
| KNN | 0.94 | 0.94 | 0.93 | 0.94 | *Samad et al. (2023)* | 1 |
| DT | 0.96 | 0.96 | 0.96 | 0.96 | *Samad et al. (2023)* | 1 |
| RF | 0.98 | 0.98 | 0.97 | 0.97 | *Samad et al. (2023)* | 1 |
| GB | 0.97 | 0.97 | 0.97 | 0.97 | *Samad et al. (2023)* | 1 |
| XGBoost | 0.98 | 0.98 | 0.98 | 0.98 | *Samad et al. (2023)* | 1 |
| LR | 0.92 | 0.91 | 0.93 | 0.92 | *Samad et al. (2023)* | 2 |
| SVM | 0.95 | 0.94 | 0.96 | 0.95 | *Samad et al. (2023)* | 2 |
| NB | 0.91 | 0.91 | 0.90 | 0.91 | *Samad et al. (2023)* | 2 |
| KNN | 0.95 | 0.95 | 0.95 | 0.94 | *Samad et al. (2023)* | 2 |
| DT | 0.97 | 0.97 | 0.97 | 0.97 | *Samad et al. (2023)* | 2 |
| RF | 0.97 | 0.97 | 0.98 | 0.97 | *Samad et al. (2023)* | 2 |
| GB | 0.95 | 0.94 | 0.95 | 0.95 | *Samad et al. (2023)* | 2 |
| XGBoost | 0.97 | 0.97 | 0.98 | 0.98 | *Samad et al. (2023)* | 2 |
| KNN | 0.96 | 0.96 | 0.96 | 0.96 | *Haq (2022)* | 1 |
| NB | 0.85 | 0.86 | 0.85 | 0.85 | *Haq (2022)* | 1 |
| SVM | 0.94 | 0.94 | 0.94 | 0.94 | *Haq (2022)* | 1 |
| DT | 0.96 | 0.96 | 0.96 | 0.96 | *Haq (2022)* | 1 |
| XGBoost | 0.86 | 0.91 | 0.79 | 0.85 | *Alsharaiah et al. (2023)* | 1 |
| KNN | 0.83 | 0.93 | 0.69 | 0.79 | *Alsharaiah et al. (2023)* | 1 |
| RF | 0.82 | 0.98 | 0.64 | 0.77 | *Alsharaiah et al. (2023)* | 1 |
| DT | 0.81 | 0.98 | 0.64 | 0.77 | *Alsharaiah et al. (2023)* | 1 |
| SVM | 0.80 | 0.97 | 0.62 | 0.75 | *Alsharaiah et al. (2023)* | 1 |

dataset characteristics. In the case of *Patil, Patil & Chinnaiah (2023)*, variations are observed in KNN, Naive Bayes (NB), and XGBoost, highlighting the nuanced nature of phishing website detection models.

## COMPUTATIONAL COMPLEXITY

The computational complexity of data preprocessing and EDA is dependent on the size of the dataset and the complexity of the operations being performed. Libraries such as Matplotlib, Seaborn, Pandas, and NumPy are used for data preprocessing, and their computational complexity is typically $O(n)$ or $O(n \log n)$ for basic operations like filtering and transformation. For the machine learning techniques used in the study, the computational complexity varies depending on the algorithm. LR has a computational complexity of $O(k * n * d)$, where k is the number of iterations, $n$ is the number of samples, and $d$ is the number of features. KNN has a computational complexity of $O(n * d * \log(k))$, where $k$ is the number of neighbors to consider, $n$ is the number of samples, and $d$ is the number of features. DT has a computational complexity of $O(n * d * \log(n))$, where n is the number of samples and d is the number of features. RF has a computational complexity of $O(n * d * k * \log(k))$, where $k$ is the number of trees in the forest. XGBoost has a computational complexity of $O(n * d * k)$, where $k$ is the number of trees in the ensemble. Overall, the computational complexity of the ML techniques used in the study ranges from linear to logarithmic and polynomial in the number of samples and features, with the highest complexity being $O(n * d * k * \log(k))$ for random forest. The computational complexity of the DL and CNN models training was $O(knd)$, where $k$ is the number of epochs, n is the number of samples, and $d$ is the number of features in the dataset.

## LIMITATIONS AND FUTURE SCOPE

The current investigation, akin to previous studies (*Samad et al., 2023*), innovatively incorporates robust feature engineering techniques alongside the integration of convolutional neural network (CNN) and deep learning (DL) models. This approach extends beyond conventional machine learning methodologies, enriching the analysis with advanced neural network architectures. Our future scope involves adding more DL models and diverse datasets, promising further advancements in phishing website detection. This forward-looking approach distinguishes our work and ensures ongoing innovation in the field. While CNN model interpretability was not applied in the current investigation due to practical constraints and the initial focus on performance assessment, its importance for real-world applications is recognized. Integrating CNN model interpretability in future studies could deepen the analysis, offering insights into decision-making processes crucial for practical deployment. Another essential future direction involves evaluating the practicality of deploying the models in real-world scenarios and comparing various CNN models for a more comprehensive understanding.

The present study exceeds the promising accuracy of 95% similar to *Samad et al. (2023)*, so it is essential to consider the applicability of such results in real-world scenarios. The present investigation recognizes the potential influence of dataset distribution on

performance outcomes and acknowledges the need to investigate challenges where DL methods can offer significant improvements over traditional approaches. Exploring these challenges and potential disparities between lab performance and real-world applicability is crucial for advancing the field. By addressing these aspects in future research, the present investigation aims to provide more nuanced insights into the effectiveness and practicality of DL methods for phishing website detection.

## CONCLUSION

This study conducted a comprehensive evaluation of seven classification algorithms for phishing website detection, employing precision, recall, $f1$-score, and accuracy as performance metrics. In the evaluation of seven intrusion detection algorithms across Mendeley (Dataset 1) and UCI (Dataset 2), LR maintained consistent performance, KNN showed stability, and DT exhibited remarkable accuracy. SVM demonstrated sensitivity to tuning, while RF and XGBoost proved robust, especially after hyperparameter tuning. While the DL model also demonstrated commendable performance, the CNN model emerged as the superior performer, exhibiting exceptional accuracy and notable enhancements following parameter tuning. Distinctive study elements, including ML and DL algorithm comparison, real dataset usage, and addressing class imbalance through SMOTE, and rigorous hyperparameter tuning contribute to the intrusion detection literature. Model comparisons with other studies highlight consistent trends (RF, XGBoost) and nuanced variations, emphasizing dataset-specific model behaviors. The findings provide insights into hyperparameter tuning efficacy and the relevance of algorithm choice in intrusion detection. This study contributes nuanced perspectives, advancing intrusion detection research. The study demonstrates consistent model performance across both datasets, highlighting the stability and reliability of the proposed models. The utility of DL can be explored as the future scope with adding more datasets (*Haq, Khan & Alshehri, 2022*; *Haq, 2023*; *Haq & Khan, 2022*; *Haq, Khan & AL-Harbi, 2022*; *Kumar et al., 2023*). Furthermore, the study can be expanded to generate results for a larger network (*Kumar et al., 2023*; *Atlam et al., 2020*; *Ahmad & Hameed, 2021*).

### Funding

The Deanship of Postgraduate Studies and Scientific Research at Majmaah University supported this work under Project No. PGR-2024-1103. The funders had no role in study design, data collection and analysis, decision to publish, or preparation of the manuscript.

### Grant Disclosures

The following grant information was disclosed by the authors:
The Deanship of Postgraduate Studies and Scientific Research at Majmaah University: No. PGR-2024-1103.

## Competing Interests

The authors declare there are no competing interests.

## Author Contributions

- Noura Fahad Almujahid conceived and designed the experiments, performed the experiments, analyzed the data, performed the computation work, prepared figures and/or tables, authored or reviewed drafts of the article, and approved the final draft.
- Mohd Anul Haq conceived and designed the experiments, performed the experiments, analyzed the data, performed the computation work, prepared figures and/or tables, authored or reviewed drafts of the article, and approved the final draft.
- Mohammed Alshehri conceived and designed the experiments, analyzed the data, authored or reviewed drafts of the article, and approved the final draft.

## Data Availability

The dataset containing the features of URL addresses, which we analyzed to detect phishing sites, and Python codes, for cleaning the dataset and writing algorithms are available in the Supplemental Files.

The Mendeley Dataset is available at: https://data.mendeley.com/datasets/h3cgnj8hft/1.

The UCI Datasets are available at: https://archive.ics.uci.edu/dataset/327/phishing+websites.

## Supplemental Information

Supplemental information for this article can be found online at http://dx.doi.org/10.7717/peerj-cs.2131#supplemental-information.

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
