# Peer review of "Comparative evaluation of machine learning algorithms for phishing site detection"

_PeerJ Computer Science, doi:10.7717/peerj-cs.2131_

## Round 0.1 · original submission · Major Revisions

· Academic Editor

Major Revisions

This review highlights critical issues in the paper's presentation, experimental design, and findings' validity. While the paper addresses phishing site detection with clarity, it needs to better emphasize its contributions in comparison to existing research. To enhance novelty, considering additional datasets and deep learning models is suggested. The conclusion should better emphasize contributions, and formatting issues should be resolved for clarity.

**Language Note:** The review process has identified that the English language must be improved. PeerJ can provide language editing services - please contact us at [email protected] for pricing (be sure to provide your manuscript number and title). Alternatively, you should make your own arrangements to improve the language quality and provide details in your response letter. – PeerJ Staff

Reviewer 1 ·

Basic reporting

This paper focuses on an important problem: phishing site detection. The authors emphasize the growing prevalence of phishing and evaluate the effectiveness of various machine-learning (ML) algorithms on phishing site detection, including Support Vector Machine (SVM), K-Nearest Neighbors (KNN), Random Forest (RF), Decision Tree (DT), Extreme Gradient Boosting (XGBoost), and Logistic Regression (LR). The results shows that ML models are effective on Mendeley dataset.
The overall writing of this paper is fluent and easy to understand. However, this paper suffers from several critical issues, I will discuss these as follows.

Experimental design

Novelty: The authors may need to highlight their contributions when compared with some existing work, especially: Analysis of the Performance Impact of Fine-Tuned Machine Learning Model for Phishing URL Detection. This prior work uses two datasets (including the Mendeley one) and also compares the performance of eight machine learning models on Phishing URL detection. Besides, it explores how tuning factors (such as dataset balancing, hyper-parameter tuning and feature selection), can influence the efficiency of machine learning algorithms. This prior work seems to cover more perspectives than this paper. I would recommend the authors to reevaluate their paper and emphasize the novelties compared to prior works in the paper. Especially in line 294: Comparison with other studies, I expect that there are some discussion how this work is better than prior work.

If the authors want to add more novelty to this paper, the authors could consider incorporating a broader array of datasets and deep learning (DL) models. This expansion would not only diversify the experimental scope but also enable a comparative evaluation of varying models across distinct datasets. Besides, if the models show different performance in different situations, the authors can explore the reasons behind and conduct an in-depth analysis of the factors influencing the performance of these models under various scenarios. Uncovering these dynamics could yield insightful observations, potentially shaping strategic decisions regarding the optimal application of specific models in different scenarios. Exploring these perspectives could lead to valuable insights, helping to make better choices about when and where to use different models most effectively.

Validity of the findings

As discussed in 2. Experimental design, the impact and novelty are not addressed clearly, which should be highlight in this paper, especially the introduction section.

Additional comments

1. The figures used in this paper are not clear(especially figure 2), and should be redrawn using appropriate tools for better visibility and understanding.

2. Please fix some type or format issues in this paper. For example, in line 151-156, the text appears to be entirely bolded, you should revise these lines to match the font style used in the rest of the document. This would involve removing the bold formatting to ensure uniformity and readability throughout the paper.

Cite this review as

Reviewer 2 ·

Basic reporting

The paper's presentation requires significant improvements and reformat for enhanced clarity and readability. The paper used the same font size and style across section titles, particularly for the "Address Bar-based Features" section, which makes it hard to tell the hierarchical differentiation for the readers. The manuscript contains excessively long paragraphs, with some exceeding a page, and lacks necessary subtitles (e.g., in the "Related Work" section). Tables and figures are placed at the end rather than integrated within the text, hampering the flow of information. Several figures are of extremely low resolution, making them unreadable, and the tables are improperly labeled (all as Table 1), with seemingly random bolding in rows. The manuscript is riddled with grammatical errors and lacks standard formatting practices.

Many key concepts and background information have remained unexplained, poorly explained, or uninterpretable. For example:
1. Why are lines 153-156 bolded, and sections in lines 151 and 152 are left empty?
2. The feature descriptions in lines 153-161 are unclear, particularly their relation to the features in Tables 2 and 3. It is suggested to use concrete examples while explaining these features. And please use consistent terms. For example, the 'pop-up window' feature is not referenced in any table.
3. On line 169, you mentioned "Analyzing correlations is a crucial aspect of EDA". Why is it important and what does such a correlation indicate?



Improvement Suggestions:
Please address the above-mentioned problems and conduct a thorough grammar check.

Experimental design

The paper's scope is limited to five basic machine learning methods (e.g., random forest) applied to a SINGLE dataset, despite mentioning many approaches and datasets in the "Related Work" section. I wonder why these state-of-the-art approaches and datasets are not evaluated. Any particular reason for choosing this particular dataset?


Improvement Suggestions:
- Explain the experimental design or extend the experiments as suggested.

Validity of the findings

The paper only studies one dataset with three existing studies ([27-29]) and concludes that the random forest method outperforms others. The validity of the findings and conclusions is questionable due to the limited scope of the study.

In addition, it is not clear why the authors chose to use this particular dataset, and whether the chosen dataset is representative or widely accepted in the area.

Questions on the comparative experiment with [27-29]:
a) Are these numbers of [27-29] copied from their paper or reproduced?
b) If it is reproduced, how much difference are the numbers compared with the original performance reported in these papers?
c) Is the same dataset and split used? etc.




Improvement Suggestions:
- Clarify the comparison experiments with [27-29].
- Conduct more rigorous evaluation of more existing datasets and state-of-the-art approaches to have meaningful conclusions.

Additional comments

While the paper's focus on machine learning methods in phishing detection is commendable, its narrow scope, poor writing, and unexplained experimental designs make the conclusions of the paper hardly meaningful.

Cite this review as

Reviewer 3 ·

Basic reporting

The article presents a straightforward structure, incorporating machine learning methods inspired by existing works. While the innovation is somewhat limited, the experiments are well-founded.

Experimental design

The experimental methods closely follow existing approaches. It is suggested that the authors provide more detailed descriptions of the training and testing procedures, including the ratio of training to testing data and whether the reported results are averages or from k-fold cross-validation.

Additionally, there is a need for comprehensive explanations and annotations for mathematical formula parameters and subscripts/superscripts, such as beta, K, and X in Formula 2.

When presenting individual experiment results, consider incorporating a summarizing statement at the beginning or end of each paragraph to facilitate readers' understanding.

Validity of the findings

Appreciation is extended for providing data and code. However, a revision of the conclusion is recommended to enhance English writing. The concluding section should emphasize your contributions and the significance of your work. Clearly state what you have achieved and its implications.

Additional comments

There is an issue with the formatting in lines 153-156 where the main text is unnecessarily bolded. Please review and correct this formatting inconsistency.

Cite this review as

---

## Round 0.2 · Major Revisions

· Academic Editor

Major Revisions

The paper presents a comparative study of machine learning and deep learning algorithms for phishing site detection using two datasets. However, reviewers highlighted several areas for improvement regarding the clarity of the paper's contributions and methodology. Specifically, concerns were raised about the lack of depth in explaining the integration of SMOTE for data imbalance mitigation and the need for a more detailed comparative analysis of model performance metrics. Furthermore, suggestions were made to enhance the experimental design by providing additional details on dataset sizes and investigating the robustness of the CNN model under reduced training data conditions. While the methods used were deemed standard and reproducible, clarity in describing evaluation metrics was recommended to improve the validity and transparency of the findings. Addressing these recommendations, along with improvements in writing style and format, would enhance the overall quality and readability of the manuscript.

**Language Note:** The review process has identified that the English language must be improved. PeerJ can provide language editing services - please contact us at [email protected] for pricing (be sure to provide your manuscript number and title). Alternatively, you should make your own arrangements to improve the language quality and provide details in your response letter. – PeerJ Staff

Reviewer 1 ·

Basic reporting

This paper compares six machine learning and one deep learning algorithms for phishing site detection, and use two datasets for evaluation. However, I still cannot get the contributions of this paper:

1. Utilization of SMOTE for Data Imbalance:
The paper mentions the application of Synthetic Minority Over-sampling Technique (SMOTE) to mitigate issues related to data imbalance. However, the description and integration of SMOTE within the study's methodology lack depth and specificity. It remains unclear whether the authors have implemented any modifications or improvements to the standard SMOTE technique or have merely employed it in its original form. Clarifying whether the implementation of SMOTE directly influences the comparative performance of the analyzed algorithms would also be beneficial.

2. Comparative Analysis and Contribution Clarity:
The authors add comparison datasets, and compare both machine learning models and deep learning models, and get the results that CNNs achieve stable and good results. However, the findings—that CNNs exhibit stable and satisfactory performance—while relevant, do not appear to offer novel insights or actionable guidance for researchers and practitioners in the phishing site detection domain. The results, as currently presented, may not sufficiently advance the understanding of how or why certain models outperform others in this specific application. To amplify the paper's impact, it would be advantageous to delve deeper into the analysis, perhaps by identifying unique characteristics of the datasets that favor CNNs, exploring model interpretability, or examining the feasibility of model deployment in real-world scenarios. Providing a more granular analysis of the models' performance, including false positive rates, detection latency, or robustness to novel phishing techniques, could render the findings more informative and compelling. Also, there are many different CNN models worth comparing.

Recommendations for Improvement:

1. Enhance the comparative analysis by offering deeper insights into the models' performance metrics and their implications for phishing site detection.
3. Consider discussing the practical applicability of the findings, including model deployment challenges, to offer tangible guidance to the field.

Experimental design

See the comments above.

Validity of the findings

See the comments above.

Additional comments

See the comments above.

Cite this review as

Reviewer 3 ·

Basic reporting

The manuscript exhibits persistent issues with its format and writing style. It is recommended that the author enhance their writing skills and address the formatting concerns. Additionally, the paper lacks details on the experimental content, necessitating an expansion of the experimental section.

Experimental design

While the inclusion of a Convolutional Neural Network (CNN) in addition to traditional machine learning methods enhances the comprehensiveness of the evaluation, the paper lacks essential details on the experimental setup. The author should provide additional information on the size of the training and testing datasets. Furthermore, given the superior performance of CNN, it is suggested to investigate if such performance holds with a reduced training dataset. This analysis would contribute valuable insights into the model's robustness under varying data conditions.

Validity of the findings

The chosen methods are straightforward and commonly used, ensuring reproducibility. However, the paper could benefit from a more explicit description of the evaluation metrics used and their appropriateness for phishing site detection. Providing clarity on these aspects would enhance the overall validity and transparency of the findings.

Additional comments

1. The manuscript requires improvements in writing style and format to enhance overall readability.
2. The author is encouraged to elaborate on experimental details, including the size of datasets used for training and testing.
3. Consider exploring the robustness of the CNN model under reduced training data conditions for a more comprehensive analysis.
4. Ensure clarity in describing the evaluation metrics used for a more transparent interpretation of findings.

Cite this review as

---

## Round 0.3 · Major Revisions

· Academic Editor

Major Revisions

The manuscript exhibits notable improvements in presentation, yet lingering issues with table and figure positioning and labeling persist. Clarity is needed regarding the integration of these elements within the text and rectifying inconsistencies in table captions. Despite commendable additions such as a new dataset and evaluation method (CNN), the rationale behind dataset selection remains obscure, and discrepancies in findings compared to prior research raise concerns. Queries regarding differences in results between studies using the same method and articulation of the manuscript's novelty are raised, along with concerns about high performance and dataset specificity. Formatting issues impact readability and professionalism, with suggestions for transitioning to LaTeX proposed. The exclusion of contemporary methods like K-means raises concerns about study comprehensiveness, prompting recommendations for expanding the methodological framework. Enhanced transparency regarding hyperparameter tuning methodology would improve reproducibility and rigor. Overall, while improvements are acknowledged, addressing these concerns is essential for enhancing the manuscript's quality and impact.

Reviewer 2 ·

Basic reporting

Compared to the last submission, there are some major improvements to the presentation. But the position of tables and figures are still not fixed. I wonder if is there any specific reason why the tables and figures cannot be positioned in the body of the paper along with the text, but rather they are all put after the content. In addition, all tables are captioned as "Table 1". Please fix.

Experimental design

Compared to the last submission, the authors added a new dataset and a new method (CNN) for evaluation. However, some details regarding the selection of the datasets are still missing. There are many relevant datasets, as mentioned in related work, such as [20] and [23], but the paper chose two. Please specify any reasons the authors chose these two datasets. Are they the most representative, latest, or largest?

Validity of the findings

In Table 8, if using the same method, i.e., LR, why are the results of the current study and [27] different?

Additional comments

Reviewer 2's comments did not seem to be responded to in the response provided.
I can see some improvements in both presentation and experimental designs compared to the last submission. However, the novelty or contribution of this paper is still not clear. This paper seems very similar to the existing work [27], as they are both empirical studies for various ML methods on phishing website detection. If I understand correctly, the current work simply extends [27] on one DL method (CNN). Applying SMOTE (an existing method) to an existing dataset contributed little to the novelty. Please specify the difference between the current work and [27].

If the paper aims to be an extension of [27] on DL methods, the paper may want to focus on DL methods instead of reproducing many studied methods (Random forest, KNN, etc.), and more DL methods should be evaluated. However, given the high performance achieved by the previous works (over 95%), there is not much space for improvements if stick with the current dataset and problem setup. I wonder is the performance really this good in real-world applications? Could the good performance be because of the specific dataset distribution? There may be some challenges to this problem that need to be discovered, where DL methods can significantly outperform traditional methods.

Cite this review as

Reviewer 3 ·

Basic reporting

The manuscript still presents formatting issues, such as inconsistent paragraph indentation and line spacing within sections. These aspects are critical for readability and professional presentation; hence, the authors are urged to prioritize rectifying these discrepancies. Transitioning to LaTeX may streamline the formatting process and enhance the overall presentation of the manuscript.

Moreover, while the manuscript addresses the comparative evaluation of machine learning algorithms, it primarily focuses on traditional approaches, omitting more contemporary methods like K-means. This oversight undermines the comprehensive nature of the study and warrants acknowledgment from the authors to manage reader expectations effectively.

Experimental design

The reliance solely on conventional machine learning techniques raises concerns regarding the comprehensiveness and relevance of the experimental design. Given the rapidly evolving landscape of machine learning, incorporating a broader spectrum of algorithms, including both traditional and contemporary ones, would enhance the study's robustness and relevance. The authors are encouraged to consider expanding their methodological framework to encompass a more diverse range of machine learning approaches.

Validity of the findings

Hyperparameter tuning plays a pivotal role in optimizing the performance of machine learning models, particularly traditional ones. Providing insights into the methodology employed for hyperparameter selection would enhance the transparency and reproducibility of the findings. Therefore, the authors are encouraged to elaborate on their approach to hyperparameter tuning, including any strategies or algorithms utilized to optimize model performance effectively.

Additional comments

See the comments above.

Cite this review as

---

## Round 0.4 · Minor Revisions

· Academic Editor

Minor Revisions

The initial reviewer expressed concerns about the manuscript's lack of innovation and significant overlap with previous studies.

One reviewer questioned the choice of CNN over DNN and urged clarification on dataset usage to address concerns about potential overfitting.

The reviewer emphasized the need for further enhancements in experimental diversity and feature engineering to elevate the manuscript's quality and impact.

Reviewer 2 ·

Basic reporting

The layout of Tables 7 and 8 could be improved for clarity and straightforward comparison. I recommend organizing comparable results either in the same row or in adjacent rows to facilitate easier comparison. For instance, group all four rows of RL results together in Table 7.

Experimental design

I find the authors' response to the question "There are many relevant datasets, as mentioned in related work, such as [20] and [23], but the paper chose two. Please specify any reasons the authors chose these two datasets. Are they the most representative, latest, or largest" unclear and unspecific. Please specify with more details including a list of datasets that are available, and why EXACTLY they are excluded or why these two datasets are chosen.

Validity of the findings

no comment

Cite this review as

Reviewer 3 ·

Basic reporting

Thank you for your revisions and improvements to the manuscript. I appreciate the efforts made to address the previous concerns. However, I have some additional comments and suggestions that I believe could further enhance the quality and impact of your work:
1. While the introduction of the Convolutional Neural Network (CNN) is a step towards enhancing the manuscript's innovation, the level of novelty still appears to be insufficient, with too many overlaps with previous studies [27]. To strengthen the innovative aspect of your research, it would be beneficial to consider the inclusion of additional deep learning techniques for evaluation, rather than limiting the analysis to just CNN.
2. Although increasing the experimental scale can partially compensate for the lack of innovation, the current study seems to fall short in comparison to earlier works. The datasets and testing scenarios are too limited, and the evaluation methods are not as comprehensive as they could be. I encourage you to diversify your experimental settings to demonstrate the robustness and applicability of your proposed methods across a wider range of contexts.
3. Feature selection and engineering play a critical role in the performance of machine learning models. Have the authors considered further refining the selected features through filtering, recombination, or dimensionality reduction techniques to observe potential improvements in experimental performance? Moreover, designing new features could significantly enhance model performance and contribute to the overall innovation of the paper.
In summary, while the authors have made progress in revising the manuscript, there is still room for improvement in terms of innovation, experimental diversity, and feature engineering. I trust that the suggestions provided will guide you in refining your research and enhancing the manuscript's contribution to the field.

Experimental design

See above comments

Validity of the findings

See above comments

Additional comments

See above comments

Cite this review as

Reviewer 4 ·

Basic reporting

1. Please do not use "EdrawMind" format for your rebuttal letter.
2. The paper is well-written with professional English. The technical terms are appropriately explained, making it easier for readers to understand.
3.The background section effectively sets up the importance of phishing detection and the advancements in machine learning that facilitate it. The literature review is comprehensive, though integrating more recent references could further enhance it.
4. In the rebuttal letter, comments are well solved.

Experimental design

The paper clearly defines its objective to evaluate multiple machine learning algorithms for phishing detection, which is both relevant and significant within the cybersecurity field.
The methods are thoroughly described, providing enough detail for reproducibility. The use of SMOTE to address class imbalance is particularly noteworthy, showcasing a rigorous approach to data handling.
The research fits within the journal's scope and demonstrates a rigorous approach by employing a variety of machine learning models, ensuring the study's robustness.

Validity of the findings

The statistical analysis and machine learning techniques are appropriate for the research questions. The comparative evaluation of the models is well-executed, with clear tables and figures to summarize the findings.
The conclusions are directly linked to the initial research questions and are supported by the results. The paper discusses the superior performance of CNN models in phishing detection, providing a valuable insight into their applicability in cybersecurity.

Cite this review as

Reviewer 5 ·

Basic reporting

The manuscript presents a comparative study of phishing attack detection using several kinds of ML and DL methods. The text is now matched to the figures. The manuscript is overall easy to read. However, some broken sentences or logic need fixing, e.g. lines 42-44, 46-48, and 155-158.

Experimental design

I think a fully connected deep neural network (DNN) should do the work. Why do the authors choose CNN instead of DNN as the DL method in this work?

Validity of the findings

The authors should mention how they use the datasets. How many cases are used as training sets, test sets, and validation sets? Will the DL method overfit?

Cite this review as

---

## Round 0.5 · accepted · Accept

· Academic Editor

Accept

Congratulations! The paper meets the standards for publication. Despite the minor issues mentioned, the paper's strengths in methodology and results warrant its acceptance.

Reviewer 2 ·

Basic reporting

See Additional comments

Experimental design

See Additional comments

Validity of the findings

See Additional comments

Additional comments

My review from the last round is not addressed or responded:


Basic reporting

The layout of Tables 7 and 8 could be improved for clarity and straightforward comparison. I recommend organizing comparable results either in the same row or in adjacent rows to facilitate easier comparison. For instance, group all four rows of RL results together in Table 7.


Experimental design

I find the authors' response to the question "There are many relevant datasets, as mentioned in related work, such as [20] and [23], but the paper chose two. Please specify any reasons the authors chose these two datasets. Are they the most representative, latest, or largest" unclear and unspecific. Please specify with more details including a list of datasets that are available, and why EXACTLY they are excluded or why these two datasets are chosen.

Cite this review as

Reviewer 4 ·

Basic reporting

The rebuttal answered my questions well.

Experimental design

no comment

Validity of the findings

no comment

Cite this review as

Reviewer 5 ·

Basic reporting

I appreciate the efforts made by the author and the manuscript has improved greatly. A comparison with DNN and more explanations of feature selection is added to the manuscript. I am satisfied with the change.

Experimental design

I see no problem with the experimental design.

Validity of the findings

Adding the comparison with DNN makes the findings more solid so I am satisfied with the change.

Cite this review as